behaviour/biomechanics/neuroscience

human locomotion, kinematic coordination, synergy, biomechanical constraints

**Author for correspondence:**
Caihua Xiong
e-mail: chxiong@hust.edu.cn

# Common kinematic synergies of various human locomotor behaviours

Bo Huang, Caihua Xiong, Wenbin Chen, Jiejunyi Liang, Bai-Yang Sun and Xuan Gong

Institute of Robotics Research (IR2), State Key Laboratory of Digital Manufacturing Equipment and Technology, Huazhong University of Science and Technology, Wuhan, Hubei 430074, People's Republic of China

BH, 0000-0001-7168-4495; CX, 0000-0003-2326-0289; B-YS, 0000-0001-5714-4812

Humans show a variety of locomotor behaviours in daily living, varying in locomotor modes and interaction styles with the external environment. However, how this excellent motor ability is formed, whether there are some invariants underlying various locomotor behaviours and simplifying their generation, and what factors contribute to the invariants remain unclear. Here, we find three common kinematic synergies that form the six joint motions of one lower limb during walking, running, hopping and sitting-down-standing-up (movement variance accounted for greater than 90%), through identifying the coordination characteristics of 36 lower limb motor tasks in diverse environments. This finding supports the notion that humans simplify the generation of various motor behaviours through re-using several basic motor modules, rather than developing entirely new modules for each behaviour. Moreover, a potential link is also found between these synergies and the unique biomechanical characteristics of the human musculoskeletal system (muscular-articular connective architecture and bone shape), and the patterns of inter-joint coordination are consistent with the energy-saving mechanism in locomotion by using biarticular muscles as efficient mechanical energy transducers between joints. Altogether, our work helps understand the formation mechanisms of human locomotion from a holistic viewpoint and evokes inspirations for the development of artificial limbs imitating human motor ability.

## 1. Introduction

Humans demonstrate excellent locomotor ability in daily living. For example, they can walk, run or hop in an effortless way [1,2]. However, why and how they can produce various locomotor behaviours effortlessly, a feat that many of us take for granted, is

still poorly understood [3,4]. The complex and redundant human musculoskeletal system, including 31 bones, three major joints (the hip, knee and ankle joints) and more than 50 muscles in each lower limb [5], endows humans with enough flexibility to produce various locomotor behaviours. However, such complexity and redundancy also bring challenges about motor control. The central nervous system (CNS) always has to choose among multiple possible solutions for a given behavioural goal and then coordinates the many degrees of freedom of the musculoskeletal system [6]. In addition, in response to diverse environmental constraints, the CNS also needs to make adaptive adjustments, which further complicates the motor control [7].

In the last few decades, researchers have realized that there are certain principles in human locomotion. The many degrees of freedom in the lower limb are not independent, but constrained by the nervous system [8]. Specifically, a number of studies suggest that the degrees of freedom problem can be solved by a modular control architecture [4,9,10]. The CNS simplifies the control of complex motor behaviours by combining a small number of primitive motor modules [10–12], where each module is a functional unit that specifies a coordination pattern of multiple degrees of freedom. In this context, motor modules at different levels (often called synergies) have been revealed, thereby providing support for the modular control hypothesis [10]. For example, studies on the neural control of muscles show that a few muscle synergies sufficiently describe various muscle activation patterns in human locomotion [4,13,14]. Studies on kinematics find that the joint motions of the lower limb can be effectively explained by a few kinematic synergies in some typical human movements (e.g. squats, walking and going up or down a step) [15]. The temporal changes of the elevation angles of lower limb segments are also found to be covariant along an attractor planar in human walking, running and hopping, referred to as planar covariation law [12,16–19]. Taken together, these motor synergies suggest an effective strategy for the CNS to produce complex motor behaviours in a simple manner. However, a fundamental question regarding the generation of human locomotion still remains unclear: do there exist some invariant and common synergies underlying various human locomotor behaviours and simplifying their generation?

Generally, it is often assumed that different locomotor behaviours require the CNS to develop behaviour-specific synergies to meet some unique motor requirements, such as executing specific limb endpoint motion [12] or limiting the energy cost of locomotion [8]. Although this adaptability in motor control gives humans the ability to find the best way to achieve a given behavioural goal, it still seems to be an impossible or daunting task for humans to produce various locomotor behaviours because of the diversity and complexity of behaviours. In this context, it can be predicted that humans can further simplify the generation of various human locomotor tasks through re-using some common and basic synergies (if they exist), without the need to develop entirely new synergies for each task.

In addition, another fundamental question regarding the synergies is to elucidate where they originate from [10,20]. Generally, neural control and biomechanical constraints from the musculoskeletal system are considered as two main factors that contribute to limb coordination [20,21]. However, it is difficult to reveal and distinguish their respective contributions due to complex interactions. Here, it can be expected that the common synergies, if they exist, are associated with the biomechanical constraints, because both of them have consistent effects on different motor tasks and describe inherent limb movement characteristics in a similar way [20]. However, it remains unclear which biomechanical constraints are responsible for the synergies and what effect each constraint has on the generation of limb movements.

In the present study, we investigated whether some kinematic synergies were conserved across various human locomotor behaviours to simplify their generation. To this end, we characterized the kinematic coordination of the human lower limb during 36 different motor tasks, including walking, running, hopping, turning and sitting-down-standing-up, while considering different environmental conditions. These motor tasks to some extent represent the versatile motor ability of the lower limb in daily living [22,23]. Consistent with previous studies [18,24], principal component analysis (PCA) was applied to extract the common kinematic synergies (CKSs), and detailed coordination relationships among joint motions were assessed by correlation analysis. Next, in combination with the biomechanical constraints from the musculoskeletal system, we further tried to establish a link between the common coordination characteristics and the biomechanical characteristics of the human lower limb in order to find corresponding evidence for the CKSs. We hypothesized that some invariant and basic kinematic synergies underlie the generation of human locomotion, and various complex locomotor behaviours can be effectively constructed by their combination. We predicted that there would be biomechanical support for the CKSs and the effect of the biomechanical constraints on the generation of limb movements would also be uncovered by their link with limb coordination characteristics.

# 2. Material and methods

## 2.1. Experimental protocol

Nine healthy male participants (age: $23.0 \pm 1.0$ years; height: $173.1 \pm 4.1$ cm; mass: $64.0 \pm 6.1$ kg; mean ± s.d.) with no history of gait impairments participated in the study. The Chinese Ethics Committee of Registering Clinical Trails approved the experimental protocol and all participants gave written informed consent before the experiment.

To collect motion data from motor tasks which can represent the versatile motor ability of the human lower limb [22,23], five basic motor modes were selected: walking (Nos. 1–15; figure 1), running (Nos. 18–28), hopping (Nos. 31–36), turning (Nos. 29 and 30) and sitting-down-standing-up (Nos. 16 and 17). Moreover, considering the effect of natural environment constraints, the participants were asked to walk or run under different ground conditions: level ground (Nos. 7 and 22), cross slopes (incline angle: ± 14.5°; Nos. 1, 2, 18 and 19), longitudinal slopes (incline angle: ± 2.6° and ± 6°; Nos. 3, 4, 12, 13, 20, 21, 27 and 28), obstacles (width: 30 cm; height: 10 or 20 cm; Nos. 8–11 and Nos. 23–26) and stairs (riser: 15 cm; tread: 30 cm; Nos. 5, 6, 14 and 15). Except for walking and running, other motor modes were performed on level ground. Detailed description for these motor tasks is available in the electronic supplementary material.

Finally, the motion data from 36 different motor tasks were collected. For all the tasks, the participants were instructed to move in a natural way, so that they chose their preferred speeds and cadences (detailed gait parameters are reported in electronic supplementary material, table S1). Moreover, all the tasks were recorded three times after a few practices and in a random order.

Human kinematic data were recorded at 100 Hz by using the Vicon Motion Capture System (Oxford Metrics, UK) with 10 cameras. Twenty reflective markers (diameter: 14 mm) were attached to the body landmarks of both lower limbs according to the Plug-in Gait model provided by Nexus software (Oxford Metrics, UK). In addition, two additional calibration markers were attached to the left and right medial malleoli during a static trial (standing still). During hopping, ground reaction forces were recorded at 1000 Hz by using four AMTI force plates ($60 \times 40$ cm; Advanced Mechanical Technology Inc., USA) placed at the centre of the walkway and along the motion direction.

## 2.2. Data pre-processing

Marker trajectories were filtered by a Woltring filter with a mean-squared error of 20 mm². The ground reaction forces were low-pass filtered with a fourth-order Butterworth filter (cut-off frequency: 25 Hz). Kinematic data (i.e. joint angles) were calculated by the Plug-in Gait model.

For walking and running, the timing of foot contact was defined as the timing when the heel marker became nearly stationary (speed less than $0.4$ m s$^{-1}$) [25]. For walking or running on level ground, on cross slopes, on longitudinal slopes and over an obstacle, we retained the motion data over only a gait cycle for each trial according to two successive foot contacts. For hopping, the foot contact was determined by the timing when the vertical ground reaction force was more than 7% of the body weight. The motion data during a middle gait cycle of each hopping trial were retained for further analysis. For walking on the stairs, turning and sitting-down-standing-up, the motion data from the movement beginning time to the completion time were retained for each trial. A detailed description is available in the electronic supplementary material. After selecting the main motion data, the joint angle sequences for each trial were resampled to 200 points using cubic spline interpolation.

## 2.3. Extraction of kinematic synergies

Here, the movement relationships among six joint motions in the lower limb were studied: hip flexion/extension (H f/e), hip adduction/abduction (H a/a), hip internal/external rotation (H rot), knee flexion/extension (K f/e), ankle plantarflexion/dorsiflexion (A p/d) and ankle internal/external rotation (A rot; i.e. adduction/abduction [5]). Flexion, adduction, internal rotation and dorsiflexion were defined as positive values, and the posture during the static trial was set as initial posture (the mean joint angles during the static trial were subtracted from the joint angle values during dynamic trials). As in many previous studies [12,17,18], here we characterized the joint coordination for each lower limb (left or right lower limb) separately in this study and did not examine the joint coordination between both lower limbs. To extract task-specific synergies for each lower limb in each task, the motion data from

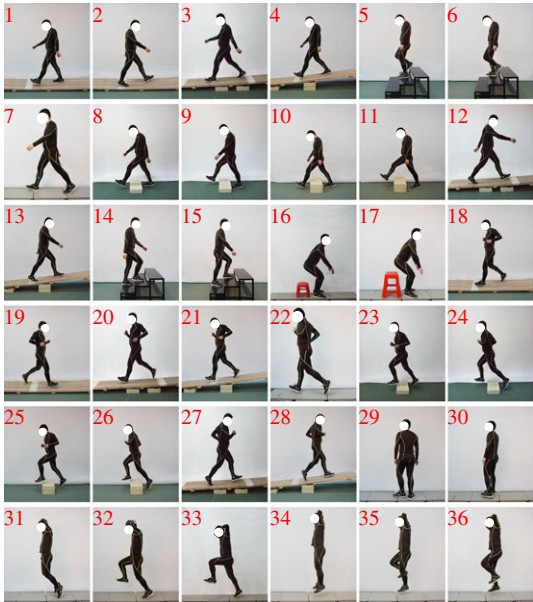

**Figure 1.** Experimental tasks: 36 different motor tasks were included in this study.

the same task (e.g. $j$th task) were pooled together as a posture matrix for each participant,

$$\mathbf{Q}_j = [\,\mathbf{q}_1 \quad \cdots \quad \mathbf{q}_i \quad \cdots \quad \mathbf{q}_{600}\,] \in \mathfrak{R}^{6\times600},$$

where $\mathbf{q}_i = [q_{1i}\, q_{2i} \ldots q_{6i}]^T \in \mathfrak{R}^{6\times1}$ represents corresponding six joint angle values of the left or right lower limb at the $i$th moment. Then to extract the CKSs for each lower limb individually, the motion data from all 36 tasks were pooled together as an entire posture matrix $\mathbf{Q}_e \in \mathfrak{R}^{6\times21600}$ for the left or right lower limb of each participant.

PCA was employed to identify the kinematic synergies, consistent with previous studies [18,24]. By using PCA on the posture matrix $\mathbf{Q}_e$ or $\mathbf{Q}_j$, the CKSs for all motor tasks or task-specific synergies for the $j$th task were extracted, and original joint motions could be reconstructed by the linear combination of several synergies,

$$\mathbf{q}_i - \bar{\mathbf{q}}_i = \sum_{k=1}^{6} c_{ki}\mathbf{s}_k,$$

where $\bar{\mathbf{q}}_i$ is the average of $\mathbf{q}_i$, $\mathbf{s}_k$ is the $k$th synergy equal to the direction of the $k$th principal component (the eigenvector of the covariance matrix of joint motions with the $k$th largest eigenvalue), the elements of $\mathbf{s}_k$ (weightings) represent the contributions of joint motions to the synergy (the absolute value of a weighting above 0.25 was defined as indicating significant contribution [26]), and $c_{ki}$ (recruitment coefficient of synergy) represents the contribution of the $k$th synergy to the original joint motions at the $i$th moment. The number of synergies needed to be retained was defined as the minimum number of synergies that could account for more than 90% of the total variance [27].

## 2.4. Coordination relationships among joint motions

The essence for the existence of the CKSs is the coordination existing among joint motions. Therefore, we further studied whether there were some coordinated joint motions during performing these rich motor tasks to verify the rationality for the CKSs. Coordinated joint motions should be synchronous; thus, the absolute value of the correlation coefficient was used to measure the degree of the coordination relationship between each pair of joint motions while ignoring their relative motion direction [21,24]. A higher correlation coefficient means a stronger coordination relationship. Then by using it as a similarity measure, the agglomerative hierarchical clustering method was used to study and show the coordination relationships among joint motions in a more intuitive way. Before the start of clustering, the absolute values of the correlation coefficients were averaged across all participants. After the start, the similarity between two clusters was determined by the average linkage algorithm [28].

It is known that humans coordinate lower limb degrees of freedom in a task-specific way for each different motor task. Accordingly, different common coordination relationship may be extracted from the motion data consisting of different tasks. Then one question is raised: are there invariable or basic common coordination characteristics within the lower limbs when more and more tasks are performed? To address this question and further verify the rationality of the CKSs, we explored the relationship between limb coordination with the number of performed tasks. In the case of $n$ performed tasks, a general index, called mean coordination relationship, was used to evaluate the common coordination relationship and defined as the average of the correlation coefficients across the combination possibilities of joint motion pairs ($C_6^2 = 15$ possibilities) and the motion subsets composed of $n$ tasks ($C_{36}^n$ or 2000 possibilities; further details in electronic supplementary material).

## 2.5. Similarity of synergies

The similarity between two synergies was assessed using the absolute value of their scalar product. Two synergies were considered significantly similar if their similarity was greater than 0.7 [29,30]. Using this similarity measure, we compared the CKSs between the left and right lower limbs and among participants (the similarities between all pairs of participants) and investigated a basic question of the extent to which the CKSs were associated with the task-specific synergies for each single task.

## 2.6. Movement reconstruction

To further verify the ability of the CKSs to characterize each motor task, here we assessed the reconstruction accuracy of each task by the CKSs. As described above, when the CKSs were extracted from all motor tasks of each participant, the recruitment coefficients of the CKSs were also obtained for the movement process of each task. Therefore, the joint motions in a task could be reconstructed by the CKSs and their recruitment coefficients in the task. Finally, the percentage of the movement variance of the task which was explained by the CKSs was used to evaluate the quality of movement reconstruction: the sum of the variance of reconstructed joint motions divided by the sum of the variance of original joint motions in the task.

## 2.7. Statistics

The clustering result based on the averages of the correlation coefficients across all participants might differ from the results based on individual data, due to individual differences. Thus, we tested the statistical significance of the final clustering result. In each agglomerative step, the two most similar clusters were merged. We used a one-tailed paired $t$-test or a one-tailed Wilcoxon signed-rank test (when the sample data were not normally distributed according to Lilliefors test) to separately test whether the similarity between the two merged clusters was significantly higher than the similarities between both of them with each of the other existing clusters when clustering happened. The $p$-value was defined as the maximum of multiple comparisons and was adjusted by Bonferroni correction. Moreover, the differences in the similarities between two pairs of synergies and in the contributions of the CKSs to the reconstruction of joint motions were also analysed by one-tailed paired $t$-tests or one-tailed Wilcoxon signed-rank tests. The statistical tests were performed using MATLAB (MathWorks) and the significance level was set at $\alpha = 0.05$.

# 3. Results

## 3.1. Existence and characteristics of the common kinematic synergies

In this section, we verified the existence of the CKSs conserved in diverse motor behaviours and revealed their basic characteristics. The results showed that the abundant natural movements of the human lower limb could be effectively constructed by the combination of a small number of CKSs. Most of the total variance in the motion data of all the 36 motor tasks could be explained by only the first three synergies (greater than 90%; $95.23 \pm 1.29\%$ and $96.84 \pm 0.61\%$ in the left and right legs, respectively; figure 2a). The finding indicated that independent variables with a lower dimensionality effectively described lower limb movements even when humans needed to perform a large number of motor tasks and thus provided preliminary support for the existence of the CKSs.

The weightings of the distribution of the CKSs showed basic coordination characteristics in the lower limb (figure 2). Overall, the weightings of the three primary CKSs were consistent between left and right

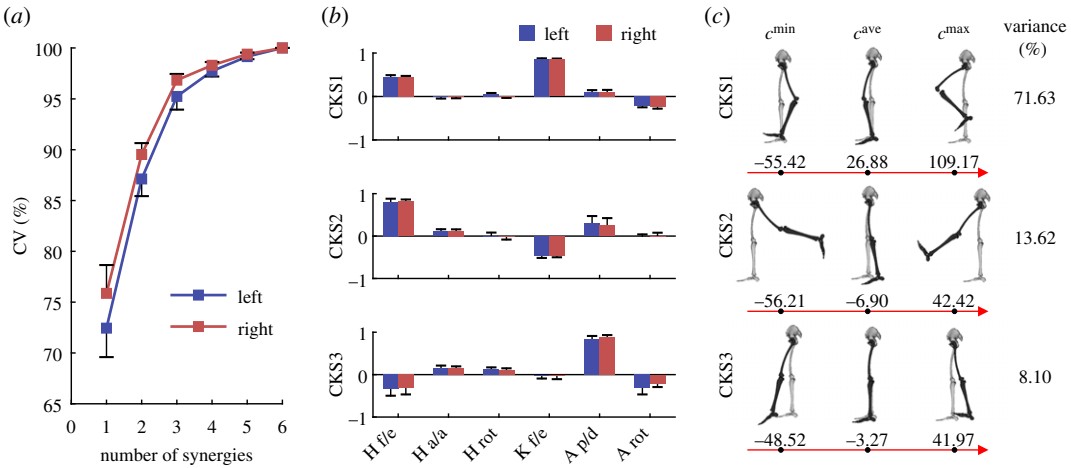

**Figure 2.** There do exist some CKSs in human lower limb. (*a*) Cumulative percentage of the variance (CV) in the motion data of all the 36 motor tasks explained by the CKSs (means and standard deviations across participants). The CKSs are extracted from all the 36 motor tasks for each participant, and the first three CKSs account for most of the variance. (*b*) The main three CKSs for the left (blue) and right (red) lower limbs. Abbreviations: H f/e, hip flexion/extension; H a/a, hip adduction/abduction; H rot, hip internal/external rotation; K f/e, knee flexion/extension; A p/d, ankle plantarflexion/dorsiflexion; A rot, ankle internal/external rotation. (*c*) Eigen-movements along the three CKSs extracted from the motion data of both lower limbs of all participants. [$c^{min}$,$c^{max}$] is the motion range along the CKSs; $c^{ave}$ is the average between $c^{min}$ and $c^{max}$. The variance represents the percentage of the total variance in the motion data of all tasks of both lower limbs and all participants explained by corresponding CKS. The skeletal model is adapted from an existing model [31–34] and demonstrated by OpenSim software [35].

legs (similarity = 0.996 ± 0.002, 0.98 ± 0.03, 0.97 ± 0.04 for the first, second and third CKSs, respectively) and among individuals (left leg: similarity = 0.99 ± 0.004, 0.95 ± 0.04, 0.93 ± 0.05; right leg: similarity = 0.99 ± 0.003, 0.96 ± 0.05, 0.95 ± 0.05) (see also electronic supplementary material, figure S1). Based on this consistency, we further extracted the CKSs from the motion data of both lower limbs of all participants (the motion data were pooled together as an entire data matrix, like the data of the same task of a participant) and showed the eigen-movements along them (figure 2*c*), which described the overall coordination characteristics for both lower limbs and all participants. To be specific, the first CKS was mainly characterized by in-phase coordination of the hip and knee: hip-and-knee flexion and hip-and-knee extension (the absolute values of their weightings greater than 0.25; figure 2*b*). And ankle rotation had a significant weighting (average = 0.251) in this synergy of the right leg. By contrast, the second CKS consisted of the coordinated movements among hip flexion, knee extension and ankle dorsiflexion, and among hip extension, knee flexion and ankle plantarflexion. The first two CKSs seem to be complementary and allow more motor behaviours to be produced. Finally, the third CKS consisted of the coordinated movements between hip flexion and ankle plantarflexion, and between hip extension and ankle dorsiflexion. And ankle rotation showed a noticeable contribution to this synergy of the left leg (average weighting = 0.32). Moreover, ankle plantarflexion/dorsiflexion had the highest weighting in the third CKS (figure 2*b*), indicating the importance of ankle joint motions in the generation of lower limb movements.

Consistent with the first CKS, the clustering analyses of joint motions showed that there was a main coordinated subgroup in the lower limb consisting of hip flexion/extension, knee flexion/extension and ankle rotation (*p* = 0.012; figure 3*a*; *p* = 0.012; figure 3*b*). Then, similar to the characteristics of the second and third CKSs, ankle plantarflexion/dorsiflexion also had a relatively weaker coordination relationship with the subgroup. By contrast, hip abduction/adduction and rotation were relatively independent of the other joint motions (figure 3). Altogether, humans did coordinate joint motions in achieving diverse motor tasks, indicating that the CKSs are not just the result from matrix factorization (i.e. PCA), but represent some basic coordination characteristics within the human lower limb.

## 3.2. Invariance of the common coordination characteristics

Humans adopted task-specific strategies for different motor tasks, which could be appreciated by the differences in the coordination characteristics of different tasks (electronic supplementary material, figure S2). In line with this observation, the coordination relationships among joint motions were also

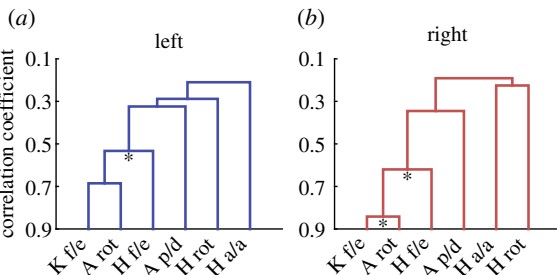

**Figure 3.** Dendrogram of the coordination relationships among joint motions in the left (*a*) and right (*b*) lower limbs. A lower node in the dendrograms indicates a stronger correlation between corresponding joint motion subgroups. The asterisks under nodes indicate the statistical significance of corresponding clustering results (*$p < 0.05$).

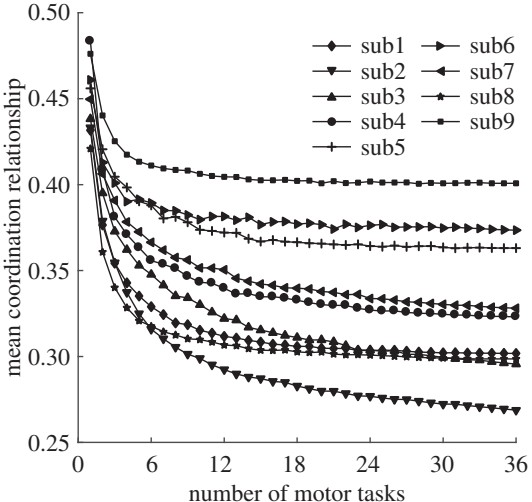

**Figure 4.** Invariance of the common coordination characteristics. The mean coordination relationship (left leg) does not always decrease significantly with the increase of tasks and will be nearly stable, especially when more than half of all tasks are included.

strongest when only one task was performed as expected and decreased quickly with the increase of tasks despite individual differences (figure 4; see also electronic supplementary material for the right leg, electronic supplementary material, figure S3). However, the common coordination relationship did not always decrease significantly and would be nearly invariable, especially when more than half of all tasks were selected. This invariance thus implied that there were indeed some invariant coordination laws governing the generation of lower limb movements and representing some intrinsic control strategies for humans to perform these rich motor behaviours.

## 3.3. Ability of the common kinematic synergies to characterize each motor task

The movement reconstruction of each motor task by the CKSs showed that the three CKSs could describe the movement characteristics of walking, sitting-down-standing-up, running and hopping (average reconstruction accuracy greater than 90%, ranging from $90.72 \pm 2.89\%$ to $99.10 \pm 0.52\%$ in both legs), except for a small number of motor tasks: turning in place (Nos. 29 and 30), the movement of the free leg (in the air all the time) during one-legged hopping (left leg: Nos. 33 and 36; figure 5*a*; right leg: Nos. 32 and 35; figure 5*b*) and hopping forward on only the left leg (No. 32; figure 5*a*). In particular, the contributions of the three CKSs showed some differences in different motor tasks and some tasks could be effectively described by only part of the three CKSs (reconstruction accuracy greater than 90%). Sitting-down-standing-up only required the first CKS (Nos. 16 and 17). Walking upstairs could be reconstructed by the first (No. 14; figure 5*a*; No. 15; figure 5*b*) or the first two (No. 15; figure 5*a*; No. 14; figure 5*b*) CKSs. Likewise, the joint motions could also be reconstructed by the first two CKSs in the other five tasks: walking over a 20 cm obstacle (right leg: Nos. 10 and 11; figure 5*b*) and the movement of trailing limb during running over an obstacle (left leg: No. 26; figure 5*a*; right leg: Nos.

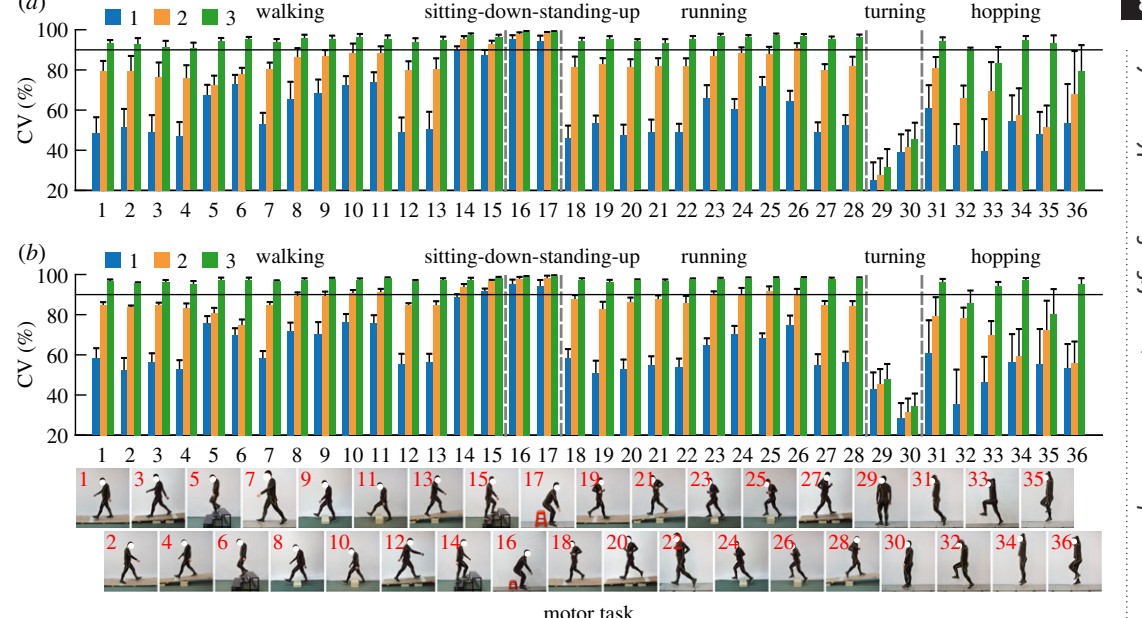

**Figure 5.** Movement reconstruction of each task by the combination of the CKSs. The percentage of the total joint motion variance of each task (CV; means and standard deviations across participants) explained by the first (blue), the first two (orange) and the first three (green) CKSs in the left (*a*) and right (*b*) lower limbs. The reconstruction accuracy is considered good if CV > 90% (black horizontal lines).

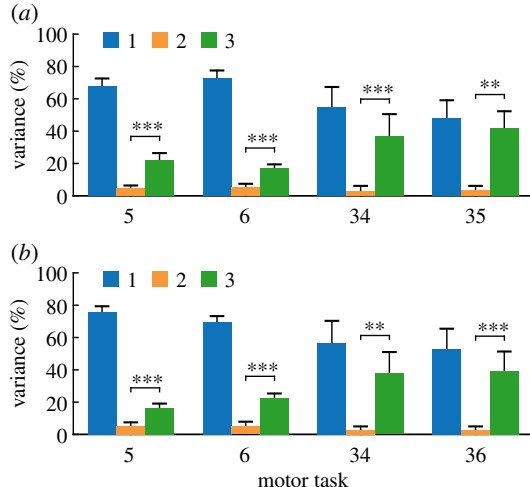

**Figure 6.** The percentage of the total joint motion variance explained by the first (blue), second (orange) and third (green) CKSs in eight specific motor tasks of left (*a*) and right (*b*) lower limbs. In these tasks, the variance of joint motions can be explained more by the third CKS than the second one (** $p < 0.01$, *** $p < 0.001$).

23 and 25; figure 5*b*). Moreover, in walking downstairs (Nos. 5 and 6; figure 5) and hopping in place (Nos. 34 and 35; figure 5*a*; Nos. 34 and 36; figure 5*b*), the variance of joint motions was explained more by the third CKS than the second one (all *p*-values less than 0.01; figure 6). And it was found that walking downstairs and hopping in place could be well reconstructed by only the first and third CKSs (reconstruction accuracy ranged from 90.17 ± 3.93% to 94.16 ± 2.37%), except for the left leg of walking downstairs (Nos. 5 and 6; figure 6*a*; reconstruction accuracy less than 90%).

The comparison between the CKSs and task-specific synergies of each task showed the robustness of the CKSs as well. Similar to the common coordination characteristics in all tasks, lower limb movements could be generated by combining one, two or three basic task-specific synergies for each task (electronic supplementary material, figure S4). Therefore, the CKSs can evolve into task-specific synergies in terms of effective degrees of freedom. By computing their similarities, we further found that the CKSs were

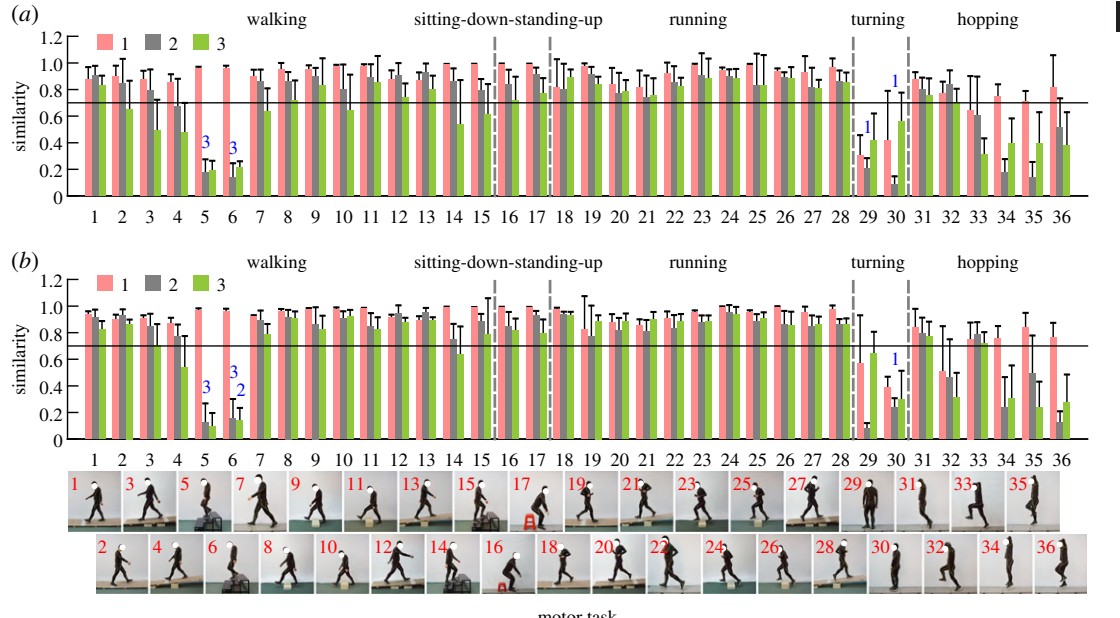

**Figure 7.** Associations between the common and task-specific synergies. The similarities (means and standard deviations across participants) between the first common and the first task-specific synergies (pink), between the second common and the second task-specific synergies (grey), and between the third common and the third task-specific synergies (green) for the left (*a*) and right (*b*) lower limbs. Two synergies are considered significantly similar if their similarity was greater than 0.7 (black horizontal lines). The blue 3 (representing the third CKS) above the grey bars of the second task-specific synergies of tasks Nos. 5 and 6 indicates that the second task-specific synergies in these tasks are more similar to the third CKS than the second CKS, and the average similarities are more than 0.7. Similarly, the blue 1 and 2 in the figure have similar meanings as the blue 3 (all *p*-values < 0.01).

consistent with the task-specific synergies of walking, sitting-down-standing-up, running and hopping forward. However, to cope with certain behaviour-specific demands, the CKSs were adjusted by humans at different levels, especially for walking downhill (Nos. 3 and 4), walking downstairs (Nos. 5 and 6), turning in place (Nos. 29 and 30), hopping in place (Nos. 34–36) and the movement of the free leg during hopping forward on one leg (left leg: No. 33; figure 7*a*; right leg: No. 32; figure 7*b*). As shown in figure 7, the first CKS was consistent with the first task-specific synergies for almost all of the tasks (average similarity greater than 0.71 in both legs), except for turning in place and the free leg during hopping forward on one leg (similarity less than 0.7). The second common and task-specific synergies were similar as well (average similarity greater than 0.74), but not including walking downhill (only in the left leg: No. 4; figure 7*a*), walking downstairs, turning, the free leg during one-legged hopping and hopping in place. For the third synergies, similar common and task-specific ones were also observed in most of tasks (average similarity greater than 0.70), except for walking on a cross slope (No. 2; figure 7*a*), walking downhill (Nos. 3 and 4; figure 7*a*; No. 4; figure 7*b*), walking downstairs, level walking (No. 7; figure 7*a*), walking over a 20 cm obstacle (No. 10; figure 7*a*), walking upstairs (Nos. 14 and 15; figure 7*a*; No. 14; figure 7*b*), turning, the free leg during one-legged hopping and hopping in place.

Moreover, by comparison, it was found that the second task-specific synergies in walking downstairs were more similar to the third CKS than the second CKS (all *p*-values < 0.01), and the average similarities were more than 0.7 (ranging from 0.93 ± 0.02 to 0.96 ± 0.05). By contrast, the third task-specific synergy in walking downstairs was similar to the second CKS (similarity = 0.78 ± 0.20; No. 6; figure 7*b*), rather than the third CKS (*p* = 0.002). Similarly, the second synergies of turning were consistent with the first CKS (Nos. 29 and 30; figure 7*a*; No. 30; figure 7*b*; similarity = 0.83 ± 0.11, 0.75 ± 0.32, 0.87 ± 0.04, respectively), rather than the second CKS (all *p*-values < 0.01).

Combining with the results of movement reconstruction (figure 5), we extracted the CKSs once again from all motor tasks except for the tasks which could not be effectively reconstructed by the CKSs. The new CKSs (electronic supplementary material, figure S5) showed coordination characteristics consistent with those illustrated in figure 2 (average similarity greater than 0.99 for the first, second and third CKSs in each leg).

# 4. Discussion

We investigated whether humans used certain common synergies to produce diverse locomotor behaviours in this study. The results provided a global perspective on understanding the formation mechanisms of human locomotion rather than a local observation on each single locomotor behaviour as before. Often, researchers have derived kinematic synergies by considering the step-to-step variability or across-subject variability for a single motor task. Here our goal was to capture the variability across tasks, instead of examining the variability within each single task. In this way, here matrix factorization and correlation analyses revealed the existence, basic characteristics and invariance of the common synergies, which indicated that various locomotor behaviours could be generated by selecting and combining several basic CKSs. Then our results further showed that the CKSs could characterize the motor ability of the lower limbs well and could evolve into the task-specific synergies by appropriate adjustments. Altogether, these CKSs suggest a basic and simple control architecture for human locomotion. Through re-using these basic synergies, humans can simplify the generation of various locomotor behaviours, without the need to develop entirely new synergies for each task.

A question remains unclear: where do the CKSs emerge from? As discussed in the following sections, biomechanical support for these synergies can be found, as suggested by a relation between the biomechanical characteristics of the musculoskeletal system and the common coordination characteristics. Meanwhile, this relation also suggests some other potential biological significances of the CKSs, in addition to serving as basic motor modules of human locomotion.

Our results suggest that hip-and-knee flexion/extension are coordinated by humans in performing various motor behaviours. This coordination relationship should be associated with the activities of the biarticular muscles crossing the hip and knee (e.g. rectus femoris and hamstrings) [36]. During hip-and-knee flexion or hip-and-knee extension movement (the first CKS; figure 2b), any one of these biarticular muscles is allowed to be fully used, functioning as an efficient force or mechanical energy transducer between joints (electronic supplementary material, figure S6). Specifically, any one of these biarticular muscles can shorten (doing positive work) at one joint (the hip or knee) and simultaneously lengthen (doing negative work) at the other joint (the knee or hip) in the above two coordinated movements. In this way, efficient muscle forces from some of these biarticular muscles can be used to flex/extend the hip and knee joints due to the small net change in muscle length (i.e. small muscle contraction velocity) and the intrinsic force–velocity relationship of muscle [37–39], and meanwhile, the mechanical energy is efficiently transferred between both joints [38,40,41]. This transfer mechanism thereby supports the existence of the first CKS and suggests that this synergy reflects an effective coordination strategy adopted by humans to save the energy cost in locomotion. The second and first CKSs are complementary. Their combination allows more motor behaviours to be produced. In addition, the coordination manners of the second synergy (hip-flexion-and-knee-extension and hip-extension-and-knee-flexion) allow the biarticular muscles to store or release large amounts of mechanical energy through simultaneous lengthening or shortening at two joints (electronic supplementary material, figure S6). This coordination strategy will be useful in the tasks requiring energy [5]. Likewise, the coordination between ankle plantarflexion/dorsiflexion and other joint motions should be associated with the biarticular muscles crossing the knee and ankle (e.g. gastrocnemius).

A locking mechanism of the knee, called 'screw-home' rotation (electronic supplementary material, figure S6), may play a role in the coordination among ankle rotation, hip flexion/extension and knee flexion/extension (figure 3). This screw-home rotation is a mechanically coupled rotation where full knee extension is always accompanied by about 10 degrees of knee external rotation due to several architectural features of the musculoskeletal system, especially the asymmetry in the shapes of the lateral and medial femoral condyles (electronic supplementary material, figure S6) [5,42]. In practice, this rotation can be generated by the external rotation of the tibia relative to the femur, which will further lead to the internal rotation of the foot (especially a constrained foot) relative to the tibia (i.e. ankle internal rotation). Therefore, ankle rotation is closely coordinated with hip-and-knee flexion/extension. Finally, in contrast with those coordinated joint motions, special hip adductors (e.g. adductor brevis), abductors (e.g. gluteus mediums), rotators (e.g. piriformis and obturator internus) [5] and their special functions (e.g. controlling frontal plane stability and achieving the rotation of the body) may lead to relatively independent hip abduction/adduction and rotation.

Altogether, our findings suggest a potential link between the CKSs and the biomechanical constraints of the musculoskeletal system. This link provides underpinnings for these synergies and facilitates our understanding of the formation mechanisms of human locomotion. Some effects of the biomechanical constraints on the generation of locomotor behaviours are revealed by this link. A basis is also provided

for further studying the interactions among the nervous system, the musculoskeletal system and the environment in human locomotion. In another aspect, it is possible that the coordination characteristics of the human lower limb are observed merely from the construction of the musculoskeletal system. However, here we characterize the basic movement laws in a quantitative way and find that humans can produce various locomotor behaviours flexibly and rapidly by combining a few common synergies.

During performing various motor tasks, humans need to follow the same constraints or achieve the same biomechanical sub-goals, which leads to the CKSs [9,15,43]. However, some specific requirements also need to be met for individual motor tasks, which may require task-specific synergies different from the CKSs. Therefore, although the CKSs describe the common coordination characteristics of human locomotion very well, they also need to be adjusted at different levels to evolve into individual task-specific synergies, especially for walking downhill, walking downstairs, turning in place, hopping in place and the free leg during one-legged hopping. Specifically, while deriving the CKSs, we have treated all 36 tasks equally. The tasks requiring specific synergies to meet their behaviour-specific goals cannot be prioritized by the PCA procedure, which may explain the large differences between their task-specific synergies and the CKSs. For instance, different from the other motor tasks, the primary goal of walking downhill and downstairs for humans is to control the descent of the body with caution [44,45]. During hopping and turning in place, humans only need to control the length change of the whole lower limb [12] or the rotation of the body with no need for expending too much effort to push the body forward. Likewise, during one-legged hopping, the function of the free leg seems to be balance control, rather than locomotion.

Muscles, as primary stabilizers and movers of the skeletal system [5], must also play an important role in the adjustments from the common to the task-specific synergies. To perform a motor task or to cope with a specific motor requirement, the muscles are excited and modulated by the CNS and sensory feedback, which subsequently produce appropriate forces to maintain or adjust the posture and movement of the target limb [39,46]. For example, during the single support phase, the stance leg is generally modelled as an inverted pendulum in walking [47] and a spring-mass system in running [48]. The differences in locomotor patterns between both gaits are related to muscle activities, which alter the stiffness distribution among joints dynamically and then lead to the differences [49–51]. Furthermore, leg stiffness is also adjusted to accommodate the surfaces of different stiffnesses when humans run and hop [52,53] and the differential stiffness distribution among joints has been demonstrated to play a role in the changes in limb coordination [12]. Collectively, these lines of evidence imply that the CKSs can evolve into various task-specific synergies by the adjustments of muscle activities under the modulation of the nervous system and sensory feedback. From this perspective, the nervous system, muscles and sensory feedback shape the motion patterns for each motor task and ultimately lead to the transition from the common to the task-specific synergies.

There are many issues about the CKSs not examined in this study. First, the PCA is a coordinate-dependent dimensionality reduction method. That is, what PCs or synergies we get will depend on the relative weighting of the different degrees of freedom (i.e. joint motions). As is conventional in previous work [18,20], here we have simply assumed equal weighting, so that all original joint motions are treated equally and the joint motions with larger amplitudes can also have a larger effect on the determination of the synergies. However, in some circumstances, more coordination characteristics in limb movements may be revealed by using re-weighted PCA (e.g. the joint motions are scaled using standard deviations or maximal active ranges of joint motions [26]). Second, the coordination between both the lower limbs is not examined in this study. This study mainly focuses on exploring the existence of the CKSs in each lower limb separately and their potential biomechanical support. In this context, what the coordination characteristics are and whether the CKSs also exist when considering both limbs simultaneously deserve to be further investigated. Third, as we do not have muscle activation information and do not perform inverse dynamics, we must treat the aforementioned relationship between the synergies and the use of biarticular muscles to be informed speculation, or a hypothesis to be tested in greater detail in a future study. Finally, the synergies are often observed at both the kinematic and muscular levels. Here we did not collect muscle EMGs, performing muscle synergy analyses with EMGs or estimated muscle forces will provide more detailed and complementary information regarding the control of movement.

## 5. Conclusion

In conclusion, our observations and analyses suggest that the combination of a few CKSs underlie a variety of complex human locomotor behaviours, consistent with the biomechanical constraints from

the musculoskeletal system and the energy-saving mechanism in locomotion. Moreover, task-specific motor demands can be met by modifying the CKSs under the regulation of the CNS.

In addition to helping us understand the motor ability of the human lower limb, our findings also have potential influences on other aspects related to the generation of limb movements. For instance, they can provide some theoretical supports for next-generation artificial limbs to reproduce human-like motor ability. The methods used in this study can provide inspiration for the exploration of the motor control mechanisms of other limbs or animals.

Ethics. The Chinese Ethics Committee of Registering Clinical Trails approved the study (permit no. ChiECRCT20200232) and the participants gave written informed consent.

Data accessibility. The motion datasets generated and analysed during the current study are available from the Dryad Digital Repository: https://doi.org/10.5061/dryad.wdbrv15n9 [54].

Authors' contributions. C.X. conceived and directed the study. B.H. and W.C. designed the experiments. B.H. collected and analysed the data. B.H., C.X., J.L., B.-Y.S. and X.G. interpreted the results and wrote the manuscript. All authors gave final approval for publication.

Competing interests. We declare we have no competing interests.

Funding. This work was partially supported by the National Natural Science Foundation of China (grant nos. 52027806, U1913601, 91648203, U1913205, 52075191 and 52005191).

Acknowledgements. We thank Longfei Cheng and Di Hu for providing valuable comments on this manuscript, and Quanlin Li for statistical advice.

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
