## [Peer Review File · Royal Society Open Science]

Review History

Decision letter (RSOS-210161.R0)

Dear Mr Huang

On behalf of the Editors, we are pleased to inform you that your Manuscript RSOS-210161 "Invariant coordination characteristics of various human locomotor behaviours" has been accepted for publication in Royal Society Open Science subject to minor revision in accordance with the referees' reports. Please find the referees' comments along with any feedback from the Editors below my signature.

Please submit your revised manuscript and required files (see below) no later than 7 days from today's (ie 15-Mar-2021) date. Note: the ScholarOne system will 'lock' if submission of the revision is attempted 7 or more days after the deadline. If you do not think you will be able to meet this deadline please contact the editorial office immediately.

on behalf of Dr Manoj Srinivasan (Associate Editor) and Kevin Padian (Subject Editor)
openscience@royalsociety.org

Associate Editor Comments to Author (Dr Manoj Srinivasan):

Associate Editor: 1

Comments to the Author:

This manuscript describes a nice useful contribution to biomechanics. The authors have addressed many of the reviewer concerns from the previous version, but I recommend making the following revisions to clarify and make the claims follow better from the analyses.

To hew close to the content of the manuscript and not introduce new words in the title, please consider renaming the title to: "Common kinematic synergies of various human locomotor behaviors". I think this is decriptive and therefore likely to be impactful.

p5. 102. The authors have used 200 points per trial, but each trial was trimmed differently, so is of different durations. In table S1 you report the completion times of the different trials for which you are not trimming. Perhaps include the one stride or step duration for the other trials too.

p5. 105-115. It is unclear what "unilateral lower limb" means. Specifically, what do you mean by "unilateral" here? I see that you do the PCA analysis for each leg separately and then compare the individual legs's PCA. Please state this explicitly using a couple of sentences in this methods section, rather than describe it as "unilateral lower limb", as the word "unilateral" is not sufficient to convey what you're doing here. "We consider each limb separately: that is, we obtain the common kinematic synergies for each limb individually (and then later compare these synergies across limbs)."

On this topic, please also explicitly note: "We did not derive CKS by considering both limbs together in this study (e.g., as in [references])". Some readers may get confused that this is what you did.

Perhaps mention in the Discussion, describing why you did not do CKS using both limbs simultaneously. [e.g., we wanted to see to what extent CKS for just a single limb is explanatory ...]

In the discussion, perhaps add a sentence like: "The PCA is a coordinate dependent dimensionality reduction. That is, what PCs we get will depend on the relative weighting of the different degrees of freedom. As is conventional in previous work [references], here we have simply assumed equal weighting, so that all joint angles are treated equally."

Potentially re-weighted PCA can address the issue of some tasks being better approximated than others.

The previous reviewers asked about how a purely kinematic approach differs from using, say, EMG. I would suggest more explicitly acknowledging this issue in the Discussion section, stating that "while we did not collect EMGs, performing such analyses with EMGs or estimated muscle forces can provide more detailed and complementary information regarding the control of movement".

Regarding your remarks about how the synergies are consistent with the efficient use of biarticular muscles, one of the previous reviewers remarked that you could not make such inferences, as you do not have muscle activation information. Please explicitly acknowledge that issue: "As we do not have muscle activation information and have not performed inverse dynamics, we must treat the aforementioned relationship between the synergies and the use of biarticular muscles to be informed speculation, or a hypothesis to be tested in greater detail in a future study." Overall, I'd encourage reducing this aspect a bit, as the paper stands on the merits of its good results already.

Another thing to acknowledge is that: "While deriving the CKS, we have treated all 36 tasks equally. Because a good majority of the tasks were symmetric tasks like walking, running, and symmetric hopping, it may be that the asymmetric tasks like turning in place were not prioritized by the PCA procedure, which may explain why turning was poorly explained the CKS (or equivalently, why the CKS were substantially different from the task-specific synergies for turning in place."

Finally, please note somewhere something to the effect of: "Often, researchers have derived kinematic synergies by considering the step to step variability or across subject variability for a single task. Here our goal was to capture variability across task, but keeping the number of steps and subjects relatively small".

===PREPARING YOUR MANUSCRIPT===

- one version identifying all the changes that have been made (for instance, in coloured highlight, in bold text, or tracked changes);
- a 'clean' version of the new manuscript that incorporates the changes made, but does not highlight them. This version will be used for typesetting.

===PREPARING YOUR REVISION IN SCHOLARONE===

<https://royalsociety.org/journals/authors/author-guidelines/#data>. You should ensure that

you cite the dataset in your reference list. If you have deposited data etc in the Dryad repository, please only include the 'For publication' link at this stage. You should remove the 'For review' link.

-- If you have uploaded ESM files, please ensure you follow the guidance at <https://royalsociety.org/journals/authors/author-guidelines/#supplementary-material> to include a suitable title and informative caption. An example of appropriate titling and captioning may be found at https://figshare.com/articles/Table_S2_from_Is_there_a_trade-off_between_peak_performance_and_performance_breadth_across_temperatures_for_aerobic_sc_ope_in_teleost_fishes_/3843624.

Author's Response to Decision Letter for (RSOS-210161.R0)

See Appendix A.

Decision letter (RSOS-210161.R1)

Dear Mr Huang,

I am pleased to inform you that your manuscript entitled "Common kinematic synergies of various human locomotor behaviours" is now accepted for publication in Royal Society Open Science.

Please see the Royal Society Publishing guidance on how you may share your accepted author manuscript at <https://royalsociety.org/journals/ethics-policies/media-embargo/>. After

publication, some additional ways to effectively promote your article can also be found here <https://royalsociety.org/blog/2020/07/promoting-your-latest-paper-and-tracking-your-results/>.

on behalf of Prof Kevin Padian (Subject Editor)
openscience@royalsociety.org

Appendix A

Responses to Referee: 1

Comment: This manuscript describes a nice useful contribution to biomechanics. The authors have addressed many of the reviewer concerns from the previous version, but I recommend making the following revisions to clarify and make the claims follow better from the analyses.

Response: Thanks for your valuable comments. We have revised the manuscript according to your comments.

Comment: To hew close to the content of the manuscript and not introduce new words in the title, please consider renaming the title to: "Common kinematic synergies of various human locomotor behaviors". I think this is decriptive and therefore likely to be impactful.

Response: We have renamed the title as your suggestion.

Comment: p5. 102. The authors have used 200 points per trial, but each trial was trimmed differently, so is of different durations. In table S1 you report the completion times of the different trials for which you are not trimming. Perhaps include the one stride or step duration for the other trials too.

Response: We have added the stride duration for the other trials in table S1.

Comment: p5. 105-115. It is unclear what "unilateral lower limb" means. Specifically, what do you mean by "unilateral" here? I see that you do the PCA analysis for each leg separately and then compare the individual legs's PCA. Please state this explicitly using a couple of sentences in this methods section, rather than describe it as "unilateral lower limb", as the word "unilateral" is not sufficient to convey what you're doing here. "We consider each limb separately: that is, we obtain the common kinematic synergies for each limb individually (and then later compare these synergies across limbs)."

On this topic, please also explicitly note: "We did not derive CKS by considering both limbs together in this study (e.g., as in [references])". Some readers may get confused that this is what you did.

Perhaps mention in the Discussion, describing why you did not do CKS using both limbs simultaneously. [e.g., we wanted to see to what extent CKS for just a single limb is explanatory ...]

Response: As your suggestion, we have added the statement “As in many previous studies [12, 17, 18], here we characterized the joint coordination for each lower limb (left or right lower limb) separately in this study and did not examine the joint coordination between both lower limbs” in the Methods section. (Page 5, Lines 109–117; Page 8, Lines 186–187). In the Discussion section, we have also added the statement describing why we did not derive the CKSs using both limbs simultaneously: “the coordination between both the lower limbs is not examined in this study. This study mainly focuses on exploring the existence of the CKSs in each lower limb separately and their potential biomechanical support. In this context, what the coordination characteristics are and whether the CKSs also exist when considering both limbs simultaneously deserve to be further investigated” (Page 13, Lines 343–346).

Comment: In the discussion, perhaps add a sentence like: "The PCA is a coordinate dependent dimensionality reduction. That is, what PCs we get will depend on the relative weighting of the different degrees of freedom. As is conventional in previous work [references], here we have simply assumed equal weighting, so that all joint angles are treated equally."

Potentially re-weighted PCA can address the issue of some tasks being better approximated than others.

Response: Thanks for your suggestion, we have added the statement in the Discussion section (Page 13, Lines 337–343).

Comment: The previous reviewers asked about how a purely kinematic approach differs from using, say, EMG. I would suggest more explicitly acknowledging this issue in the Discussion section, stating that "while we did not collect EMGs, performing such analyses with EMGs or estimated muscle forces can provide more detailed and complementary information regarding the control of movement".

Response: Thanks for your kind suggestion. We have added the statement about the issue in the Discussion section (Page 14, Lines 349–351).

Comment: Regarding your remarks about how the synergies are consistent with the efficient use of biarticular muscles, one of the previous reviewers remarked that you could not make such inferences, as you do not have muscle activation information. Please explicitly acknowledge that issue: "As we do not have muscle activation information and have not performed inverse dynamics, we must treat the aforementioned relationship between the synergies and the use of biarticular muscles to be informed speculation, or a hypothesis to be tested in greater detail in a future study." Overall, I'd encourage reducing this aspect a bit, as the paper stands on the merits of its good results already.

Response: Thanks for your kind suggestion. The issue has been explicitly acknowledged in the Discussion section (Page 13, Lines 346–349). We have also revised related discussions about the relationship between the synergies and the use of biarticular muscles in the Discussion section (Page 11, Lines 270–273, Lines 280–283; Page 12, Lines 288–289, Lines 305–307).

Comment: Another thing to acknowledge is that: "While deriving the CKS, we have treated all 36 tasks equally. Because a good majority of the tasks were symmetric tasks like walking, running, and symmetric hopping, it may be that the asymmetric tasks like turning in place were not prioritized by the PCA procedure, which may explain why turning was poorly explained the CKS (or equivalently, why the CKS were substantially different from the task-specific synergies for turning in place."

Response: Thanks for your comments. Indeed, turning in place shows differences from the other motor tasks that are explored in this study. However, asymmetry may not be its uniqueness. Because, in addition to turning, there are many other asymmetry tasks in the motor tasks we collect, such as eight types of obstacle crossing, four types of movements in cross slopes, and four types of hopping on a leg. Many of them can be well explained by the CKSs. Therefore, by comparison, the uniqueness of turning is more likely to be its specific behavioural goal, which may explain why turning is poorly explained by the CKSs. Different from the other tasks included in this study with the aim to push the body forward, the primary goal of turning is to rotate the whole body (Page 12, Lines 311–323).

Comment: Finally, please note somewhere something to the effect of: "Often, researchers have

derived kinematic synergies by considering the step to step variability or across subject variability for a single task. Here our goal was to capture variability across task, but keeping the number of steps and subjects relatively small".

Response: As your suggestion, we have added the statement in the Discussion section (Page 11, Lines 261–264).

Once again, we thank you for the time you put in reviewing our manuscript and look forward to meeting your expectations.